# Whole Exome Sequencing in Coloboma/Microphthalmia: Identification of Novel and Recurrent Variants in Seven Genes

**DOI:** 10.3390/genes12010065

**Published:** 2021-01-06

**Authors:** Patricia Haug, Samuel Koller, Jordi Maggi, Elena Lang, Silke Feil, Agnès Wlodarczyk, Luzy Bähr, Katharina Steindl, Marianne Rohrbach, Christina Gerth-Kahlert, Wolfgang Berger

**Affiliations:** 1Institute of Medical Molecular Genetics, University of Zurich, 8952 Schlieren, Switzerland; patricia.haug@uzh.ch (P.H.); koller@medmolgen.uzh.ch (S.K.); maggi@medmolgen.uzh.ch (J.M.); Elena.lang@uzh.ch (E.L.); feil@medmolgen.uzh.ch (S.F.); Agnes.Wlodarczyk@uni-duesseldorf.de (A.W.); baehr@medmolgen.uzh.ch (L.B.); 2Department of Ophthalmology, University Hospital and University of Zurich, 8091 Zurich, Switzerland; Christina.Gerth-Kahlert@usz.ch; 3Institute of Medical Genetics, University of Zurich, 8952 Schlieren, Switzerland; steindl@medgen.uzh.ch; 4Division of Metabolism and Children’s Research Centre, University Children’s Hospital Zurich, 8032 Zurich, Switzerland; Marianne.rohrbach@kispi.uzh.ch; 5Neuroscience Center Zurich (ZNZ), University and ETH Zurich, 8006 Zurich, Switzerland; 6Zurich Center for Integrative Human Physiology (ZIHP), University of Zurich, 8006 Zurich, Switzerland

**Keywords:** whole-exome sequencing, microphthalmia, coloboma, genetic screening, MAC, ocular development, anterior segment dysgenesis

## Abstract

Coloboma and microphthalmia (C/M) are related congenital eye malformations, which can cause significant visual impairment. Molecular diagnosis is challenging as the genes associated to date with C/M account for only a small percentage of cases. Overall, the genetic cause remains unknown in up to 80% of patients. High throughput DNA sequencing technologies, including whole-exome sequencing (WES), are therefore a useful and efficient tool for genetic screening and identification of new mutations and novel genes in C/M. In this study, we analyzed the DNA of 19 patients with C/M from 15 unrelated families using singleton WES and data analysis for 307 genes of interest. We identified seven novel and one recurrent potentially disease-causing variants in *CRIM1*, *CHD7*, *FAT1*, *PTCH1*, *PUF60*, *BRPF1*, and *TGFB2* in 47% of our families, three of which occurred de novo. The detection rate in patients with ocular and extraocular manifestations (67%) was higher than in patients with an isolated ocular phenotype (46%). Our study highlights the significant genetic heterogeneity in C/M cohorts and emphasizes the diagnostic power of WES for the screening of patients and families with C/M.

## 1. Introduction

Ocular coloboma and microphthalmia (C/M) are related congenital eye malformations, which form part of a continuum together with the most severe phenotype of anophthalmia [1]. Ocular coloboma refers to a segmental defect affecting all or parts of the iris, lens, choroid, retina, and optic nerve [2]. This defect is typically caused by a partial or complete failure of optic fissure closure during early eye development [2]. Two events are crucial in optic fissure closure: (i) close apposition of the edges of the optic cup and (ii) subsequent fusion of the apposed edges [3]. Disruption of one or both of these processes may result in ocular coloboma [3]. Microphthalmia is characterized by a reduced axial length of the eye [4,5]. It primarily results from reduced growth of the eye but can also occur secondary to incomplete optic fissure closure [6,7]. However, the exact pathophysiological mechanisms responsible for microphthalmia are still poorly understood [8]. Similarly, a small optic cup can also result in ocular coloboma, as close apposition of optic cup edges cannot occur [3]. It currently remains unknown whether the optic fissure defects in colobomatous microphthalmia are the cause or consequence of ocular growth defects [8]. 

C/M can be associated with severely reduced visual development or deterioration of vision through complications such as cataract or retinal detachment and accounts for approximately 15% of severe visual impairment and blindness worldwide [2,9]. Prevalence is estimated to range from 2 to 17 per 100,000 births for microphthalmia [6,7,10,11,12,13] and from 2 to 14 per 100,000 births for ocular coloboma [6,10,13,14]. C/M are genetically heterogeneous malformations with the occurrence of reduced penetrance and variable expressivity even within the same family and between the eyes of the same individual [1,3]. They can manifest bilaterally or unilaterally and occur as an isolated finding, with additional ocular and/or systemic anomalies, or as part of a recognizable syndrome [15,16]. 

Although environmental and genetic factors have been attributed to cause C/M, the majority of cases are caused by chromosomal anomalies or single gene defects [8]. To date, at least 82 genes have been associated with microphthalmia, anophthalmia, and coloboma (MAC), with each gene accounting for only a small percentage of cases [8,15,17]. Among these are genes coding for transcription factors, gene expression regulators, and proteins involved in different signaling pathways and retinoic acid metabolism [8,18]. Autosomal dominant, autosomal recessive, and X-linked dominant or recessive inheritance patterns have been associated with C/M [3,8]. Whilst a genetic cause can currently be identified in up to 80% of patients with bilateral anophthalmia or severe microphthalmia, it remains unknown in the majority of patients with other forms of MAC, particularly isolated coloboma [1]. Thus, the genetic cause of C/M remains unknown in 20–80% of patients, depending on severity, bilaterality, and presence of syndromic features [1,8]. 

The advent of high-throughput DNA sequencing technologies has led to the identification of many genes and DNA sequence variants implicated in human eye disorders and contributed to the progress in understanding the processes driving the development of the eye [2]. Next-generation sequencing (NGS) technologies are particularly efficient tools for research and diagnostic testing of genetically heterogeneous diseases, as they allow simultaneous screening of hundreds of genes [19]. In Switzerland, patients with isolated C/M rarely undergo genetic analysis, as genetic testing is not routinely covered by health insurance providers. However, disease management and genetic counseling could be improved by knowledge of the underlying genetic cause. Additionally, genetic analyses can be costly, and establishing a molecular diagnosis can be challenging due to the phenotypic variability and genetic heterogeneity of C/M [3]. An efficient screening strategy may alleviate this challenge. Since none of the numerous genes associated with C/M account for a large percentage of cases, whole-exome sequencing (WES) represents an ideal tool for genetic testing in C/M [19]. In this study, we describe the genetic analysis of 19 patients from 15 unrelated families with coloboma and/or microphthalmia using WES for all 15 index patients and data analysis for 307 genes of interest, including copy number variation (CNV) analysis.

## 2. Materials and Methods 

### 2.1. Patients

Patients with C/M were recruited from the Department of Ophthalmology at the University Hospital Zurich. Inclusion criteria were congenital anomalies, including bilateral coloboma, microphthalmia, or colobomatous microphthalmia, and/or familial and/or syndromic unilateral coloboma and/or microphthalmia. All index patients received an eye examination, including dilated fundus examination. The presence of additional ocular features or extraocular manifestations was documented. Affected family members were included in the study, and their data were obtained from clinical examinations, patient records, and direct questioning. Unaffected family members underwent ophthalmologic re-examination upon indications from segregation analysis. Blood samples were collected from all index patients, their parents, and available affected and unaffected family members. Ethical approval was obtained (Cantonal Ethics Committee of Zurich, Ref-No. 2019-00108), and written informed consent was provided by all participants or their legal guardians. The study was conducted in accordance with the principles of the Declaration of Helsinki.

### 2.2. Genes of Interest

A comprehensive list of 307 genes of interest was compiled for WES data filtering and analysis (Appendix A). Disease-associated and candidate genes were chosen from the Human Gene Mutation Database (HGMD) and literature searches. Genes associated with syndromic MAC, nonsyndromic MAC, and MAC phenotypes in animal models were included. The gene list was extended by candidate genes for MAC and candidate genes known to play a role in signaling and developmental pathways important for eye formation and function, compiled by Raca et al. [19].

### 2.3. Whole-Exome Sequencing and Data Analysis

DNA from blood samples was extracted using the Chemagic DNA Blood Kit (Perkin Elmer, Waltham, MA, USA). Singleton WES was performed for 15 index patients, as previously described [20]. Reads were aligned to the human reference genome (hg19), and variant calling was performed with BaseSpace Onsite (Illumina, San Diego, CA, USA). Alamut Batch version 1.11 (Sophia Genetics, Saint Sulpice, Switzerland) was used for variant annotation. WES data were filtered with the compiled gene list (Appendix A) in a stepwise manner. Variants with heterozygous allele frequency >1% (genome aggregation database (gnomAD) heterozygous allele frequency all populations) and homozygous allele frequency >0.00001% (gnomAD homozygous allele frequency all populations) were excluded (https://gnomad.broadinstitute.org/). The remaining variants were then filtered based on variant type and position in the gene. Missense variants were considered if predicted to affect protein function by at least three out of five in-silico prediction algorithms (Align-GVGD [21], FATHMM-MKL [22], CADD [23], MutationTaster [24], and SIFT [25]). Align-GVGD grades were interpreted according to Bergmann et al. [26] and the remaining scores according to Dong et al. [27]. Synonymous and intronic variants were considered if close to splice sites (±30 bp of the exon-intron boundary). Splice site variants with predicted splice site alterations were prioritized. In silico splicing predictions were examined using Alamut Visual version 2.10 (Sophia Genetics). Splice site variants with no predicted splice site alteration were considered in the absence of other candidate variants. Previous reports of the variant with a comparable phenotype in HGMD or literature served as an additional filter criterion. Candidate variants were classified as potentially disease-causing based on variant classification by the American College of Medical Genetics and Genomics (ACMG), reports of previous cases with a comparable phenotype, animal models, and functional studies. 

### 2.4. Segregation Analysis 

All candidate variants were confirmed by Sanger sequencing, and de novo variants were additionally confirmed by multiplex short tandem repeat (STR) analysis, which compared 24 STR markers between the index and both parents. Segregation analysis was performed for all available family members using Sanger sequencing. For PCR amplifications, the HOT FIREPol Kit (Solis Biodyne, Tartu, Estonia) was used for a total volume of 20 µL, containing 1× HOT FIREPol Buffer B2, 0.5 mM MgCl_2_, 200 µM dNTP Mix, 0.2 µM of each primer, 0.025 U/µL HOT FIREpol DNA polymerase, 10 ng of genomic DNA, and 1× S-solution in case of nonspecific amplification. Primer sequences are available upon request. A no template control (NTC) served as a negative control. PCR was performed on a Veriti 96 Well Thermal Cycler (Applied Biosystems, Waltham, MA, USA) or 2720 Thermal Cycler (Applied Biosystems) as follows: 95 °C for 15 min; 35 cycles at 95 °C for 30 s, 60 °C for 45 s, and 72 °C for 1 min; 72 °C for 10 min. PCR products were examined by agarose gel electrophoresis using 1% weight/volume (*w*/*v*) gels. Cycle sequencing was carried out with the BigDye™ Terminator v.1.1/v3.1 Cycle Sequencing Kit (Thermo Fisher Scientific, Waltham, MA, USA) in a total volume of 10 µL, containing 0.4x BigDye™ Terminator v1.1/v3.1 Ready Reaction Mix, 0.8× BigDye™ v1.1/v.3.1 Sequencing Buffer, 0.8 µM of single primer, and 0.6–1.0 µL of PCR product. PCR was performed on a Veriti 96-well Thermal Cycler (Applied Biosystems) as follows: 96 °C for 1 min; 25 cycles at 96 °C for 20 s, 58 °C for 20 s, and 60 °C for 2 min; 60 °C for 2 min. Postreaction spin column purification was carried out using Sephadex^®^ G-50 BioReagent fine (Sigma-Aldrich, St. Louis, MO, USA). Capillary electrophoresis was performed on a 3130xl Genetic Analyzer (Applied Biosystems), and sequences were visualized using Chromas version 2.6.6 (Technelysium, Brisbane, Australia). 

### 2.5. CNV Analysis and Breakpoint Assessment 

CNVs were identified based on exome coverage depth data for all 307 genes of interest using Sequence Pilot version 5.0 (JSI Medical Systems GmbH, Ettenheim, Germany). The *CRIM1* deletion was confirmed in index patient 1[III:3] from family 1 by long-range PCR and subsequent NGS sequencing of the deletion-spanning breakpoint fragment, as previously described [28]. Briefly, a fragment of 22,240 bp, including exons 14 to 17, the 3′ untranslated region (UTR) of *CRIM1*, as well as exon 9 and the 5′ UTR of *FEZ2*, was amplified with TaKaRa LA Taq polymerase (Takara Bio, Kasatsu, Japan). Long-range PCR was performed in a total volume of 30 µL, containing 1x Buffer Mg^2+^ free, 1x S-Solution (Solis Biodyne), 400 µM dNTP Mix, 0.4 µM of each primer, 0.05 U/µL LA Taq polymerase, and 50 ng of genomic DNA. PCR was carried out on a Veriti 96 Well Thermal Cycler (Applied Biosystems) as follows: 94 °C for 2 min; 35 cycles at 98 °C for 10 s and 68 °C for 12 min; 72 °C for 10 min. Primers were designed to flank the deleted region, with the forward primer (5′-AAA TGG CAC AAC CCT GAT AGC CAC ACA T-3′) located in intron 13 of *CRIM1* and the reverse primer (5′-CAC AGG GAG GGA AGC GGG GAA ATA AAA A-3′) located in intron 8 of *FEZ2*. Amplicons were fragmented using the Covaris M220 Sonicator (Covaris, Woburn, MA, USA) for a target size of 350 bp. Library preparation was carried out with the TruSeq DNA Nano Kit (Illumina) according to the manufacturer’s protocol. The library was quantified with the Qubit dsDNA High Sensitivity Kit (Thermo Fisher Scientific) and validated using the Agilent High Sensitivity DNA Kit (Agilent, Santa Clara, CA, USA) with the 2100 Bioanalyzer (Agilent), according to the manufacturer’s instructions. Final libraries were diluted to a loading concentration of 12 pM. Paired-end sequencing (2 × 151 cycles) was performed on the MiSeq system (Illumina). Data were aligned to the human reference genome (hg19) with the Burrows-Wheeler Aligner (BWA-MEM), and variant calling was performed using the Genome Analysis Toolkit (GATK). Sequences were visualized using Alamut Visual version 2.10 (Sophia Genetics). Breakpoints were identified using the NGS data and confirmed by Sanger sequencing of the junction fragments. Primer sequences are available upon request. Multiplex PCR, using primers from long-range PCR and a second forward primer (5′-GTT TCC GTT TTT GGC TTT GGC TGC TAC A-3′) located in intron 15 of *CRIM1*, was performed for segregation analysis in available family members. PCR products were analyzed by agarose gel electrophoresis and run on a 2100 Bioanalyzer (Agilent) using the Agilent DNA 12000 Assay (Agilent), according to the manufacturer’s instructions. 

The *FAT1* deletion in family 3 was confirmed by SNP array analysis of the trio (index patient and parents) using the HumanKaryomap-12 BeadChip version 1.0 (Illumina) with the REPLI-g Single Cell Kit (Qiagen, Hilden, Germany). Data were aligned to the human reference genome (hg19) and evaluated using the BlueFuse Multi software version 4.5 (Illumina). Breakpoints were mapped by analyzing WES and CHIP data. 

### 2.6. Functional Analyses by RT-PCR of Potential Splice Site Variants in CHD7, ACTG1, and EFTUD2

The effect of potential splice site variants in *CHD7*, ACTG1, and *EFTUD2* on splicing was functionally analyzed by isolating patient RNA from blood and subsequent reverse transcription PCR (RT-PCR). Whole blood was collected in PAXgene Blood RNA Tubes (PreAnalytiX, Hombrechtikon, Switzerland), and total RNA was extracted from index patients, carriers, and unaffected controls using the PAXgene Blood RNA Kit (PreAnalytiX). RNA extraction was performed according to the manufacturer’s instructions with the exception of DNaseI treatment, second elution (buffer BR5), and heat-denaturation of RNA. Extracted RNA was then treated with DNase I Amplification Grade (Invitrogen by Thermo Fisher Scientific). Subsequently, 300 or 600 ng of total RNA was reverse-transcribed into cDNA using the SuperScript III First-Synthesis SuperMix Kit (Invitrogen) with oligo(dT)20 primers, according to the manufacturer’s instructions. Part of the *CHD7* transcript containing exons 2–5, the *ACTG1* transcript containing exons 4–6, and the *EFTUD2* transcript containing exons 7–12, exons 8–11, and exons 2–27, respectively, were amplified from cDNA. PCR was performed with the HOT FIREPol Kit (Solis Biodyne) using 1 µL of cDNA on a 2720 Thermal Cycler (Applied Biosystems) as follows: 95°C for 15 min; 35 cycles at 95 °C for 1 min, 60 °C for 1 min, and 72 °C for 1.5 min; 72 °C for 10 min. Amplified products were separated by agarose gel electrophoresis using a 1% (*w*/*v*) gel. Sequences of cDNA fragments were confirmed by Sanger sequencing.

### 2.7. Functional Analysis by Minigene Assay of a Potential Splice Site Variant in EFTUD2

The effect of a potential splice site variant in *EFTUD2* on splicing was further examined in a cellular system. A minigene construct was generated according to Gamundi et al. [29] and de Heer et al. [30]. In short: the rhodopsin gene sequence spanning exons 3 to 5 was cloned into pcDNA3.1 (Invitrogen) using EcoRI and XhoI restriction sites (for primer sequences, see Gamundi et al. [29]). Rhodopsin exon 4 was excised from plasmid by PflMI and EcoNI digestion and replaced by *EFTUD2* exon 10 flanked by intronic sequences (NM_001142605.1:c.765-267 to c.889+171). Plasmid constructs (reference and variant c.765-15C>G) containing *EFTUD2* exon 10 sequence were verified by Sanger sequencing and transfected into HEK293T cells. Total RNA was then isolated and reverse transcribed into cDNA for further analysis, as previously described [20].

## 3. Results

Our study enrolled 19 patients from 15 unrelated families affected with coloboma (12/19) or colobomatous microphthalmia (7/19), as summarized in Table 1 and Table 2. Six of these patients presented with additional extraocular manifestations. Seven novel, and one recurrent, potentially disease-causing variants were identified in *CRIM1*, *CHD7*, *FAT1*, *PTCH1*, *PUF60*, *BRPF1*, and *TGFB2* in 7 out of 15 index patients (47% of the families; Table 2, Figure 1). These variants were identified in 4 out of 6 (67%) patients with ocular and extraocular manifestations and in 6 out of 13 (46%) patients with an isolated ocular phenotype (Table 1 and Table 2). All variants were classified as potentially disease-causing based on our criteria for variant filtering and interpretation. According to these filtering criteria, no conclusive disease-causing variants were identified in eight index patients and families (53%).

### 3.1. Sequence Variants and Clinical Findings in Coloboma 

Of the 12 affected patients diagnosed with coloboma, potentially disease-causing variants were identified in 7 patients (58%) from 4 families. Family 1 consisted of index patient 1[III:3] with iris coloboma in the left eye, her father (individual 1[II:3]) with iris and chorioretinal coloboma and microcornea in both eyes, and her grandfather (individual 1[I:1]) with iris coloboma and microcornea in the right eye. Disease status of the index patient’s uncle (individual 1[II:1]) and his daughter (individual 1[III:2]) was unknown as they were not available for an eye examination. According to direct questioning of the uncle, he had no ocular abnormalities, but his daughter had an abnormal pupil. Genetic analysis identified a novel heterozygous deletion in Cysteine Rich Transmembrane BMP Regulator 1 (*CRIM1*) spanning 9008 bp, including exons 15–17 and the 3′ UTR of *CRIM1* (Figure 2) in index patient 1[III:3]. Breakpoints were located based on NGS data to positions 36,769,283 and 36,778,290 of chromosome 2 (hg19), flanked by a 2-bp microhomology region (CT) (Figure 2B,C). Multiplex PCR showed segregation of the deletion with the disease, resulting in a 9612-bp fragment for the reference allele in all family members and an additional 13,232-bp fragment for the deleted allele in all affected family members and the index patient’s uncle (individual 1[II:1]) with unknown disease status (Figure 2D). The index patient’s cousin (individual 1[III:2]) was not available for genetic testing.

Index patient 2[II:1] presented with a small chorioretinal coloboma, megalocornea and posterior embryotoxon in the right eye, and optic disc coloboma in the left eye (Figure 3A–C). Furthermore, bilateral profound sensorineural hearing loss (SNHL), inner ear malformations with missing semicircular canals, enlarged aquaeductus vestibuli, audiogenic speech developmental disorder, double outlet right ventricle (DORV), ventricular septal defect (VSD), patent ductus arteriosus (PDA), pulmonary stenosis, global developmental delay, webbed neck, and dysmorphic features with plagiocephaly were documented. Genetic analysis identified a previously reported heterozygous missense variant (NM_017780.3:c.2095A>G, p.(Ser699Gly)) in Chromodomain Helicase DNA Binding Protein 7 (*CHD7*). The variant affects a highly conserved amino acid residue and is located 2-bp from the splice donor site of exon 3. The variant occurred de novo (Appendix A) and was predicted to impact protein function by three out of five in-silico prediction algorithms (MutationTaster, FATHMM-MKL, CADD). No variants matching our filtering criteria were detected in any known genes associated with Noonan syndrome, other RASopathies (*PTPN11*, *SOS1*, *RAF1*, *RIT1*, *KRAS*, *NRAS*, *BRAF*, *SHOC2*, *CBL*, *MEK1*, *MEK2*, *HRAS*, *MAP2K1*, *MAP2K2*, *NF1*), or other genes associated with CHARGE syndrome (*SEMA3E*). Thus, differential diagnoses of Noonan syndrome or phenotypically similar RASopathies are unlikely. As variant c.2095A>G in *CHD7* is located near the splice site of exon 3 and the variant has previously not been functionally analyzed, we examined its potential effect on splicing by analyzing cDNA fragments spanning exons 2–5 for index patient 2[II:1] and a control. Agarose gel electrophoresis showed an expected 871-bp fragment (Fragment 1) and four additional fragments indicating aberrant splicing for both index patient 2[II:1] and the control (Figure 4A). The 871-bp fragment (Fragment 1) corresponded to the correctly spliced transcript (Figure 4B). Sequencing of the 491-bp fragment (Fragment 2) revealed a lack of 380-nt from the 3′ end of the 431-bp exon 3, likely resulting from usage of a cryptic exonic splice donor site upstream of the canonical donor splice site of exon 3 (Figure 4B). This splice site alteration leads to a frameshift and the introduction of a premature stop codon (p.(Val573Ter)) in exon 3 (out of 38 exons) in the affected transcript. This aberrant splicing also occurred in the control but was (significantly) increased in index patient 2[II:1]. The three additional fragments corresponded to nonspecific or alternatively spliced transcripts (not sequenced).

Genetic analysis showed that index patient 3[II:1] was compound heterozygous for a novel frameshift variant (NM_005245.3:c.5970_5971del, p.(Asn1991PhefsTer19)) and a novel deletion of FAT Atypical Cadherin 1 (*FAT1*). CNV analysis and SNP array analysis revealed an approximately 1.8-Mb deletion in the 4q35.2 region, including *FAT1* and flanking genes *F11*, *MTNR1A*, and *ZFP42* (Figure 5B). Proximal and distal breakpoints were narrowed down to the positions g.(187,179,210–187,179,486 and 188,926,200–189,012,426; hg19) using WES and SNP array analysis data. According to segregation analysis and SNP array analysis, the frameshift variant was transmitted from the mother, whereas the deletion was inherited paternally (Figure 5). No variants matching our filtering criteria or CNVs were detected in *PAX6*. Index patient 3[II:1] was diagnosed with anterior polar cataract and macular hypoplasia (grade 2) [34] in both eyes, iris and chorioretinal coloboma in the right eye, left congenital ptosis, syndactyly of the 4th and 5th toe, and hearing impairment (Figure 3D–F). Both parents showed no evidence of ocular or systemic abnormalities or diseases. 

Family 4 consisted of index patient 4[II:2] with chorioretinal coloboma and Axenfeld-Rieger anomaly in both eyes, complicated by glaucoma (Figure 3G–K), her daughter (individual 4[III:1]) with iris coloboma in the right eye and iris/chorioretinal coloboma in the left eye and extraocular features (large VSD, bilateral clinodactyly V), and the index patient’s sister (individual 4[II:3]) with goniodysgenesis without glaucoma in both eyes but no coloboma. The index patient’s mother (individual 4[I:2]) presented with no ocular abnormalities but uterine fibroids and keratocystic lesions. Neurological diseases or abnormal cognitive development were not known in any of the family members. Personal history of the index patient and her mother pointed to potential additional dermatological and systemic manifestations, including basal cell nevus syndrome (BCNS), but could not be confirmed. Genetic analysis identified a novel heterozygous missense variant (NM_000264.4:c.490G>A, p.(Glu164Lys)) in Patched 1 (*PTCH1*). The variant was present in all affected individuals who presented with an abnormal iridocorneal angle (Appendix A) and was inherited from the index patient’s mother (individual 4[I:2]), who does not show goniodysgenesis or coloboma. This *PTCH1* missense variant affects a moderately conserved amino acid and was predicted to impact protein function by three out of five in-silico prediction algorithms (MutationTaster, FATHMM-MKL, CADD). Further genetic testing also identified a novel heterozygous de novo deletion in the 16p11.2 region in the index patient’s daughter (individual 4[III:1]). The deletion was approximately 585-kb in size, with breakpoints located to positions g.(29,580,020 and 30,177,240; hg19). 

### 3.2. Sequence Variants and Clinical Findings in Colobomatous Microphthalmia

Of the seven patients diagnosed with colobomatous microphthalmia, potentially disease-causing variants were identified in three index patients (43%) in three different genes (*PUF60*, *BRPF1*, and *TGFB2*). Index patient 5[II:1] presented with microphthalmia and iris coloboma in the right eye, chorioretinal coloboma in both eyes, developmental delay, short stature, and an atrial septal defect type 2 (ASD II). Genetic analysis revealed a novel heterozygous de novo frameshift variant (NM_001136033.2:c.752dup, p.(Gln252ProfsTer152)) in Poly(U) Binding Splicing Factor 60 (*PUF60*; Appendix A). CNV analysis of genes associated with microdeletion syndrome 8q24.3 (*SCRIB*, *NRBP2*, and *PUF60*) revealed a normal copy number. 

In Index patient 6[II:1], a novel heterozygous de novo frameshift variant (NM_001003694.1:c.1756_1757insT, p.(Glu586ValfsTer12)) was identified in Bromodomain And PHF Finger Containing 1 (*BRPF1*; Appendix A). Index patient 6[II:1] was diagnosed with iris and chorioretinal coloboma involving the optic disc in both eyes, associated with a small but clear lens and microphthalmia in the left eye. Cognitive and motor development appeared to be normal, considering the patient’s age and visual impairment.

Index patient 7[II:1] presented with juxtapapillary chorioretinal coloboma in the right eye and left severe microphthalmia without visual function (Figure 3L–N). Genetic screening identified a novel heterozygous missense variant (NM_001135599.3:c.1043G>A, p.(Arg348His)) in Transforming Growth Factor β 2 (*TGFB2*; Appendix A). The variant affects a highly conserved amino acid, and four in-silico prediction algorithms (SIFT, MutationTaster, FATHMM-MKL, CADD) predicted an effect on protein function. Extended examination of index patient 7[II:1] revealed aortic root enlargement. Parents were not available for segregation analysis, as index patient 7[II:1] was adopted.

### 3.3. Additional Sequence Variants

Six additional variants that met our filtering criteria but showed insufficient evidence to be classified as potentially disease-causing were identified in eight affected patients from five families, as summarized in Table 3. The missense variant in *CRIM1* was classified as an incidental finding, as the *CRIM1* deletion identified in this family was considered to be causative. The missense variant in *BRPF1* was excluded due to nonsegregation with the disease. Variants in *TBX5*, *FZD7*, and *PPP1R12A* were also dismissed due to nonsegregation with the disease and the presence of additional strong candidate variants, which were considered to be causative in the respective families. Splice site variants in *ACTG1* and *EFTUD2* were considered unlikely to be disease-causing as functional analysis showed no evidence for altered splicing (details available upon request).

## 4. Discussion

In this study, we examined 19 patients from 15 unrelated families with C/M, using WES in combination with data analysis for 307 genes of interest in order to identify disease-associated sequence alterations, including CNVs. Our screening approach identified potentially disease-causing variants in *CRIM1*, *CHD7*, *FAT1*, *PTCH1*, *PUF60*, *BRPF1*, and *TGFB2* in seven (47%) of the families. These potentially disease-causing variants occurred in genes involved in different biological processes, including chromatin remodeling and regulation (*CHD7*, *BRPF1*), cell proliferation (*PTCH1*), transcriptional regulation (*PUF60*), as well as in genes directly involved in optic fissure morphogenesis (*CRIM1*, *FAT1, TGFB2*).

### 4.1. CRIM1

An approximately 22-kb deletion, spanning exons 14 through 17 of *CRIM1*, has been previously reported in a large family with colobomatous macrophthalmia with microcornea (MACOM) syndrome [35]. MACOM syndrome is characterized by uveal coloboma, microcornea, increased axial length, severe myopia, and staphyloma [36]. Microcornea, coloboma, and variable expressivity, as seen in family 1, are characteristic findings for MACOM syndrome and consistent with the clinical features reported for individuals harboring the previously published *CRIM1* deletion [35,36]. *CRIM1* encodes a transmembrane protein with an extracellular and intracellular function [35]. It has been shown that CRIM1 binds growth factors via its cysteine-rich von Willebrand factor C domains (extracellular function) and forms complexes with ß-catenin and cadherins via its cytoplasmic domain (intracellular function) [37]. The *CRIM1* deletion in family 1 includes exons 15–17 and the 3′ UTR. The effect of this deletion on the protein level is unclear. In mice, the loss of *Crim1* results in ocular malformations similar to the anomalies seen in patients with MACOM syndrome, indicating that MACOM syndrome is caused by haploinsufficiency of *CRIM1* [35]. The *CRIM1* deletion in family 1 likely results in haploinsufficiency, which causes the observed phenotype. The intrafamilial phenotype variability seen in family 1 is not surprising, as variability in laterality and varying degrees of coloboma, microcornea, myopia, and enlarged eye size have been observed in families with MACOM syndrome [36,38]. 

### 4.2. CHD7

The protein encoded by *CHD7* is involved in the fine-tuning of gene transcription during early steps of development in various tissues via ATP-dependent remodeling of chromatin [39]. Heterozygous variants in *CHD7* are the major cause of CHARGE syndrome, which is characterized by coloboma, heart defects, atresia of choanae, retardation of growth and/or development, as well as genital and ear anomalies [26]. The clinical features apparent in index patient 2[II:1] fulfill the diagnostic criteria for CHARGE syndrome by Verloes et al. [40]. Phenotypically similar Noonan syndrome or RASopathies in general were excluded based on the absence of potentially disease-causing variants in the respective genes. The c.2095A>G missense variant identified in index patient 2[II:1] had previously been identified in patients diagnosed with CHARGE syndrome or the clinically overlapping Kallmann syndrome [26,31,32,33].

To our knowledge, this is the first study analyzing the effect of this variant on splicing. Functional analysis by RT-PCR revealed normal and aberrant splicing for index patient 2[II:1] and the control. Results showed a more pronounced partial exon 3 splicing, resulting in a lack of 380-nt from exon 3 in index patient 2[II:1] compared to the control. This suggests that the c.2095A>G missense variant weakens the canonical donor splice site of exon 3 and therefore increases the activity of a cryptic exonic donor site, while this cryptic exonic donor site in exon 3 has minimal activity in the wild-type/reference allele. The resulting mRNA is likely targeted for degradation via nonsense-mediated mRNA decay (NMD) due to a premature stop codon (p.(Val573Ter)) in exon 3 (out of 38 exons) [41]. As haploinsufficiency of *CHD7* is considered to be the pathogenic mechanism underlying CHARGE syndrome, we assume that a reduced amount of correctly spliced *CHD7* transcript, resulting from the c.2095A>G variant, is disease-causing [39]. However, quantification of normally and alternatively spliced *CHD7* transcript is needed to properly assess the effect of the c.2095A>G missense variant on splicing.

### 4.3. FAT1

Mutations in *FAT1* have been associated with glomerulotubular nephropathy, while homozygous *FAT1* frameshift variants have recently been associated with a new multisystemic disorder [42,43]. Variants in *FAT1* have further been identified in various other disorders, including multiple cancer types and patients with facioscapulohumeral dystrophy-like phenotype [44,45]. Coloboma and syndactyly present in index patient 3[II:1] are consistent with the clinical features seen in the new *FAT1*-associated multisystemic disorder, which is characterized by colobomatous microphthalmia, facial dysmorphism, ptosis, syndactyly, and occasional glomerulotubular nephropathy [43]. Renal assessment in index patient 3[II:1] was unremarkable. Cataract and hearing impairment have, so far, not been associated with this multisystemic disorder; however, it is unknown whether previously reported cases were examined for potential hearing defects. 

*FAT1* encodes an atypical cadherin and is suggested to regulate cell polarity, cell–cell adhesions, and epithelial sheet adhesion and fusion, a crucial morphogenic event during embryonic development, including optic fissure fusion [43,46]. Studies suggest that *FAT1* facilitates optic fissure fusion through epithelial cell-mediated fusion [43,47]. In support of this hypothesis, *Fat1*^−/−^ knockout mice display coloboma, microphthalmia, and anophthalmia with incomplete penetrance, whereas the eyes of *Fat1*^−/+^ mice appear normal, indicating that heterozygous *Fat1* depletion does not affect eye formation [43,48]. The causality of *FAT1* loss of function mutations and coloboma was demonstrated in zebrafish by *fat1a* knockdown and homozygous *fat1a* frameshift mutants, both of which resulted in coloboma [43]. *Fat1* knockout in mice was also shown to result in morphological defects in the lens epithelium, including disrupted columnar structure of lens epithelial cells, formation of cell aggregates in some regions with lost apical–basal polarity, and fragmented apical cell junctions, further indicating the importance of *FAT1* in eye development [48]. 

To date, *F11*, *MTNR1A,* and *ZFP42* have not been associated with hearing impairment or inner ear anomalies. However, a previously reported case with terminal deletion of chromosome 4q, corresponding to a heterozygous 6.9-Mb deletion in the 4q35.1–q35.2 region, including *FAT1*, presented with hearing impairment in addition to other features [49]. *FAT1* is, among other genes, located in the autosomal dominant deafness locus DFNA24, which has been mapped to an 8.1-Mb interval in the 4q31.1–q35.2 region [49]. Additionally, conditional *Fat1* mutant mice, with absent transmembrane domain, displayed shortening of the endolymphatic duct and the endolymphatic sac, expected to influence audition [50]. *Fat1* was further found to act synergistically with *Fat4* in cochlea morphogenesis, where defects in cochlear elongation and outer hair cell patterning in *Fat4* knockout mice were exacerbated upon heterozygous loss of *Fat1,* and cochlear dimensions were decreased upon homozygous *Fat1* loss [51]. However, further research is required to elucidate and determine the potential role of *FAT1* in hearing impairment. Absence of ocular and systemic abnormalities or diseases in the unaffected carrier father (individual 3[I:1]) is consistent with previous reports of incomplete penetrance for terminal 4q deletions [49]. The maternally transmitted *FAT1* frameshift variant (c.5970_5971del; p.(Asn1991PhefsTer19)) likely triggers NMD of the resulting mRNA. *PAX6*, which is associated with aniridia and associated ocular manifestations, including foveal hypoplasia and cataract [52], was excluded as the causative gene based on the absence of potentially disease-causing variants in this gene. Overall, we assume that the compound heterozygous frameshift variant and deletion in *FAT1* lead to the loss of FAT1 protein, which results in developmental eye anomalies such as coloboma and cataract, syndactyly, and hearing impairment, as seen in index patient 3[II:1].

### 4.4. PTCH1

Heterozygous variants in *PTCH1* are associated with BCNS (also known as Gorlin-Goltz syndrome) and holoprosencephaly (HPE), where ocular developmental abnormalities, including C/M, are classified as minor diagnostic criteria [15,53]. A frameshift variant and multiple missense variants in *PTCH1* have previously been reported in multiple cases with isolated and syndromic ocular developmental anomalies, including C/M, Peters anomaly, and Axenfeld-Rieger syndrome [54]. There were no indications for BCNS or HPE in these previously published cases. However, it is unclear whether patients were specifically examined for potential features of BCNS and HPE [55]. In family 4, personal history pointed to additional dermatological and other syndromic manifestations, including BCNS, but we have been unable to confirm these features. Interestingly, reduced penetrance with inheritance from asymptomatic parents, as seen for the *PTCH1* variant in family 4, has also been observed in three out of the six *PTCH1* missense variants identified by Chassaing et al. [54]. Ocular defects have been observed for various *PTCH1* animal models, including retinal abnormalities similar to those seen in patients with BCNS, as well as *Ptch1* mutant mice and microphthalmia upon *ptch1* suppression in zebrafish [54,56]. 

The transmembrane protein encoded by *PTCH1* is a key component of the Hedgehog (Hh) signaling pathway, in which PTCH1 acts as a receptor for Hh ligands [54,57]. Hh signaling is important for correct proliferation, differentiation, and patterning in various tissues and organs during embryogenesis [57]. PTCH1 protein thereby functions as a negative regulator by repressing downstream Hh signaling [57]. As Hh signaling plays an important role in eye morphogenesis and retinal development, and mutations in the hedgehog ligand *SHH* have been reported in patients with C/M, dysregulation of this signaling pathway caused by mutations in *PTCH1* may result in C/M and other ocular defects [15,56]. Previous functional studies have shown that Hh pathway activation is sensitive to *PTCH1* gene dosage, as decreased protein levels lead to overactivity of *SHH* signal transduction [54,56]. Although functional assays showed a deleterious effect on PTCH1 protein activity for four of the missense variants identified by Chassaing et al. [54], functional analysis is needed to confirm the deleteriousness and causality of the *PTCH1* variant (c.490G>A; p.(Glu164Lys)), identified here in family 4. 

Recurrent 16p11.2 deletions are associated with a spectrum of phenotypic abnormalities, including congenital malformations such as cardiovascular and skeletal malformations [58,59]. VSD and clinodactyly of the fifth finger, as seen in individual 4[III:1], have been previously reported for patients with 16p11.2 deletions [60,61]. Although C/M have been reported for two cases with 16p11.2 deletions and many genes within the deleted interval are expressed in the developing eye and/or retina, none of these have so far been associated with MAC or found to be important during ocular development [62]. Thus, whereas the extraocular features in individual 4[III:1] are likely attributed to the 16p11.2 deletion, it is unknown whether the bilateral coloboma also results from the 16p11.2 deletion or the *PTCH1* missense variant inherited from the coloboma-affected index patient 4[II:2] or both. As individual 4[III:1] was too young for assessment of cognitive abilities and development, the presence of additional manifestations associated with recurrent 16p11.2 deletions cannot be excluded.

### 4.5. PUF60

*PUF60* encodes a nucleic-acid-binding protein, involved in pre-mRNA splicing and regulation of transcription through interaction with other proteins [63]. Clinical and molecular data from individuals with *PUF60* variants suggest an emerging and clinically variable *PUF60*-associated syndrome, characterized by short stature, dysmorphic facial features, and structural malformations of the heart, eye, and variable other organs [64]. Index patient 5[II:1] presented with both pathognomonic (developmental delay and short stature) and variable (coloboma, microphthalmia, heart defect) manifestations seen in the proposed *PUF60* phenotype. Cranial MRI and renal assessment were unremarkable in our patient. Differential diagnosis of 8q24.3 microdeletion syndrome was excluded based on normal copy numbers of *SCRIB*, *NRBP2*, and *PUF60*. The *PUF60* frameshift variant (c.752dup; p.(Gln252ProfsTer152)) likely results in NMD of the resulting mRNA. Haploinsufficiency has been suggested to be the pathogenic mechanism underlying *PUF60* variants, with loss-of-function variants resulting in altered dosage of PUF60 isoforms and leading to altered splicing of target genes [65]. However, so far, *PUF60* does not seem to have a specific function during eye development [63]. Thus, we hypothesize that the frameshift variant in index patient 5[II:1] results in haploinsufficiency of *PUF60* and causes syndromic C/M through altered splicing of target genes.

### 4.6. BRPF1

*BRPF1* is a chromatin regulator, promoting histone acetylation by recognizing different epigenetic marks and activating histone acetyltransferases KAT6A, KAT6B, and KAT7 [66,67]. Heterozygous variants in *BRPF1* are associated with intellectual developmental disorder with dysmorphic facies and ptosis (IDDDFP), characterized by delayed psychomotor and language development, intellectual disability, and dysmorphic features [67,68]. Although ocular abnormalities have been described for IDDDFP as additional features, the occurrence of coloboma has only recently been reported for a single case with a nonsense *BRPF1* mutation [66]. Here, we report a novel de novo *BRPF1* frameshift variant associated with coloboma and previously unreported microphthalmia. Cognitive abilities and development of index patient 6[II:1] appeared normal. However, since the patient was still very young and proper assessment of development was difficult due to high impairment of vision, the presence of additional features associated with IDDDFP cannot be excluded. Monitoring the patient for the occurrence of such additional features is recommended to ensure the optimal development of the child. 

The *BRPF1* frameshift variant (c.1756_1757insT; p.(Glu586ValfsTer12)) may trigger NMD, leading to the degradation of the resulting mRNA. Interestingly, however, mutant transcript of previously reported *BRPF1* frameshift and nonsense variants predicted to trigger NMD escaped NMD partially or completely [67,68]. Rather, these mutations have been found to affect interactions of BRPF1 with KAT6A and KAT6B, thus impairing histone H3K23 acetylation [67,68]. Haploinsufficiency of *BRPF1* has been indicated as the pathogenic mechanism driving IDDDFP [68]. Complexes of BRPF1 with KAT6A and KAT6B are involved in the development of various organs, including the forebrain, and, accordingly, *Brpf1* knockout in mice leads to embryonic lethality [67]. Forebrain-specific deletion of *Brpf1* in mice further results in up- and downregulation of transcription of multiple genes, including transcription factors involved in developmental processes such as *Pax*6 [69]. These findings indicate that *BRPF1* functions as both an activator and silencer of genes [67,69]. Coloboma and microphthalmia in index patient 6[II:1] could result from *PAX6* downregulation due to reduced levels of BRPF1 protein, as haploinsufficiency of *PAX6* is associated with C/M [15,66]. Our findings support a previous association of coloboma with *BRPF1* mutations and reiterate the proposition that C/M are part of the phenotypic spectrum associated with IDDDFP.

### 4.7. TGFB2

Heterozygous mutations in *TGFB2* are associated with syndromic and nonsyndromic forms of aortic aneurysm, including Loeys-Dietz syndrome (LDS) and nonsyndromic aortic disease (NSAD) [70,71]. *TGFB2* variant (c.1042C>T; p.(Arg348Cys)), which affects the same amino acid residue as the variant identified in index patient 7[II:1], has previously been described in a family with NSAD [70]. Cardiologic examination for potential signs of LDS, prompted by our genetic findings, revealed aortic root enlargement in index patient 7[II:1]. The presence of the likely pathogenic *TGFB2* variant (c.1043G>A; p.(Arg348His)), in combination with the aortic root enlargement found in the index patient, fulfilled the criteria for the diagnosis of LDS proposed by MacCarrick et al. [72]. Thus, genetic testing led to early detection of LDS in index patient 7[II:1]. As individuals with LDS require close surveillance of the disease and timely management of existing manifestations, early detection and treatment are crucial to extending the lifespan of affected individuals [72]. Although ocular abnormalities have occasionally been reported for LDS, this is the first report of coloboma in a patient with LDS [72]. However, *Tgfb2* knockout mice were shown to exhibit numerous developmental defects, including coloboma, while *Tgfb2* inhibition via *BMP* signaling was shown to prevent optic fissure closure [73,74]. Thus, it has been proposed that *TGFB* signaling is necessary for the fusion of optic fissure margins via local induction of *BMP4* antagonists [73]. To our knowledge, this is the first report of a potentially disease-causing *TGFB2* variant in a patient with coloboma, as well as the occurrence of coloboma in LDS. However, further functional analysis is required to confirm the causality of the *TGFB2* variant (c.1043G>A; p.(Arg348His)). Our results suggest that *TGFB2* mutations in humans may result in developmental eye anomalies such as coloboma, as seen in mice. Our findings further indicate coloboma as a potential additional ocular feature of LDS. As of now, however, further research and identification of additional coloboma cases with pathogenic *TGFB2* variants are needed to elucidate the role of *TGFB2* in coloboma.

Despite the lack of a causal treatment for C/M, knowledge of the underlying genetic cause in patients is highly important for patient-oriented diagnosis and management of ocular and possibly associated extraocular abnormalities and diseases, confirmation of the clinical diagnosis, exclusion of differential diagnoses, and to guide genetic counseling [8,75]. In syndromic cases, timely disease management and early intervention are critical to ensure optimal development of the affected children [8]. Genetic testing may also provide insights into the detailed mechanisms underlying normal eye development and elucidate the cellular and molecular basis of C/M, which may prompt the development of potential therapies and lead to the identification of new causal genes [76].

The overall detection rate in our cohort was 47% (7/15 index patients). This is higher compared to the detection rates of previously published studies using direct sequencing or WES for molecular screening of multiple genes in MAC cohorts, which ranged between 11–29% [17,77,78,79,80,81]. However, it should be noted that multiple factors, including cohort size, patient inclusion criteria, previous exclusion of mutations in major MAC genes, use of additional molecular or cytogenetic methods, severity and laterality of the ocular defect, and presence of additional systemic malformations, may influence the resulting detection rate [82]. In 8 out of the 15 index patients (53%), no conclusive disease-causing variants were identified. Causal variants in these unsolved cases may be located in regions not covered by WES (deep intronic, regulatory, and large structural variants), may have been missed by our filtering strategy or falsely classified as incidental or benign, or may be located in genes not screened in this study [83].

Considering the significant genetic heterogeneity of C/M, WES represents an efficient approach for the screening of the large number of disease-associated and candidate genes, as well as for the identification of new causal genes. Furthermore, WES data can be reanalyzed for newly established C/M genes or candidate genes at a later time point without additional sequencing costs. Therefore, we propose that WES, with data analysis for disease-associated and candidate genes, is currently the most efficient and advantageous screening approach for C/M. Our findings support this approach, as potentially disease-causing variants identified in our study were predominantly found in genes implicated in single or few C/M cases and different biological processes not currently associated with C/M in humans. Direct sequencing and gene panels may still be more time- and cost-efficient than WES for certain research objectives, specific patient cohorts, or cases with clear indications for certain genes. Further, whole-genome sequencing (WGS) is another powerful and promising screening strategy as it allows for the identification of variants in coding and noncoding regions, and biases specific to other sequencing techniques can be avoided [8]. However, data interpretation is challenging due to the large number of variants in noncoding regions, and the costs of WGS are currently still higher than gene panels and WES [84]. If possible, screening approaches should also employ additional molecular or cytogenetic methods (e.g., quantitative WES data analysis or array comparative genomic hybridization (aCGH)) to detect large structural variants and chromosomal abnormalities. Irrespective of the screening approach employed, accurate and complete clinical information of patients and family members is essential for the interpretation of molecular data [84]. In certain cases, proper phenotype description may guide additional analyses, such as aCGH or RNA studies, as certain genes are associated with specific phenotypes [8].

In conclusion, we report seven novel and one recurrent potentially disease-causing variants in C/M-associated genes *CRIM1*, *CHD7*, *FAT1*, *PUF60*, *BRPF1*, and *PTCH1*, as well as the candidate gene *TGFB2*. Our findings expand the phenotype associated with *FAT1*, *BRPF1*, and *TGFB2* and implicate *TGFB2* as an additional candidate gene for coloboma. It remains to be seen if the same or similar variants will be detected in additional patients or families in the future by other groups and studies in order to reinforce the causality of the variants reported here. However, this might be less likely as C/M are rare diseases. Our study emphasizes the large genetic heterogeneity in C/M cohorts and highlights the importance of screening genes with few reported cases. Establishing a molecular diagnosis in patients with C/M remains challenging despite advances in screening technologies and continuous identification of novel causative genes due to the genetic heterogeneity and phenotypic variability associated with these ocular defects. However, WES represents a powerful and efficient screening approach for genetic testing in patients with C/M when combined with detailed clinical information and screening of a wide number of disease-associated and candidate genes.

## Figures and Tables

**Figure 1 genes-12-00065-f001:**
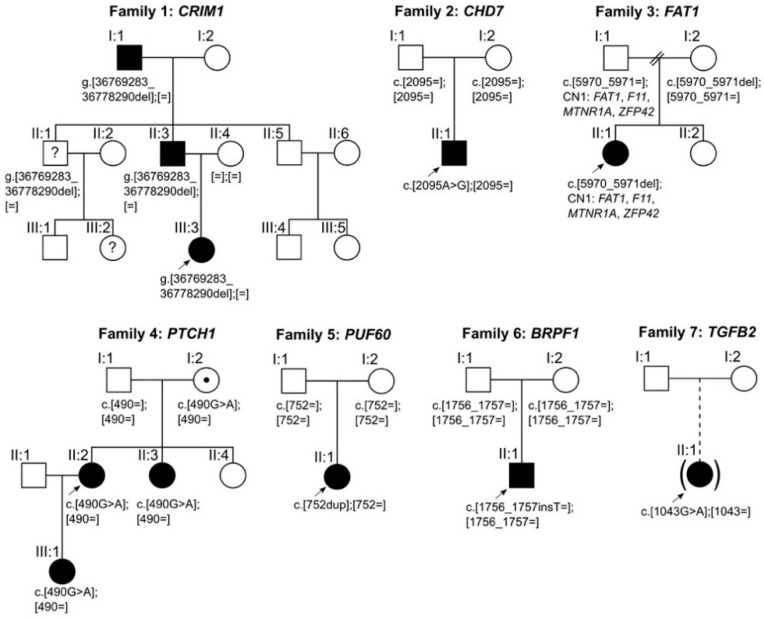
Pedigrees and segregation of potentially disease-causing variants identified in families 1–7. Squares indicate males; circles indicate females; filled symbols indicate affected individuals; symbols filled with a dot indicate asymptomatic carriers; symbols filled with a question mark indicate unknown disease status; brackets around symbols with a dashed offspring line indicate individuals adopted into the family; arrows indicate index patients. The putative disease-causing variants in families 2, 5, and 6 occurred de novo.

**Figure 2 genes-12-00065-f002:**
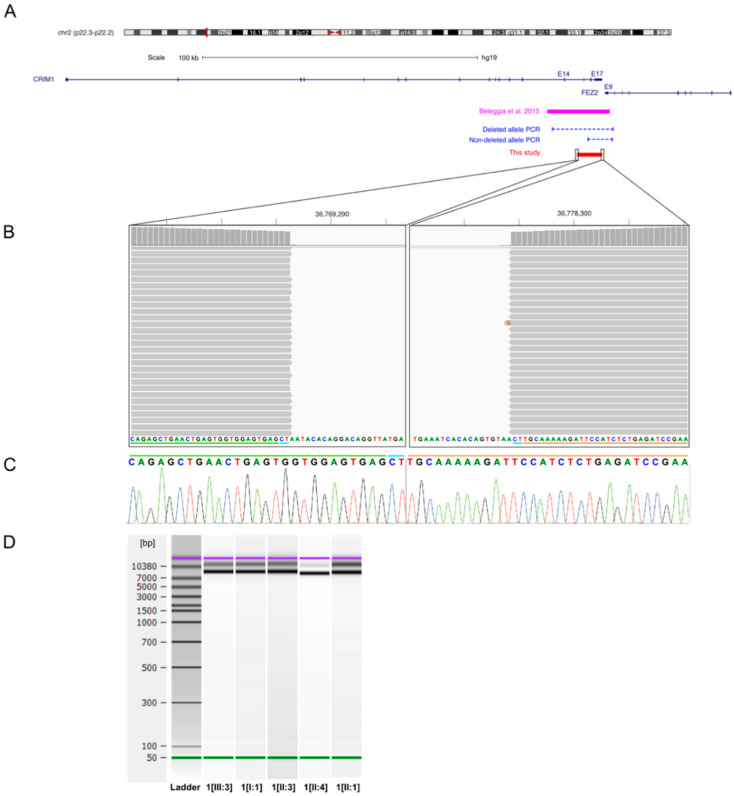
*CRIM1* deletion in index patient 1[III:3] and affected family members. (**A**) Overview of the genomic locus 2p22.3–p22.2 associated with MACOM syndrome, the previously published 22-kb *CRIM1* deletion (magenta bar), the 9008-bp deletion identified in this study (red bar), and the multiplex PCR (blue dashed lines) used for segregation analysis. The deletion in family 1 encompasses exons 15–17 and the 3′ UTR of *CRIM1*. (**B**) Alignment results of the long-range PCR product for the deleted allele of index patient 1[III:3] in the Integrative Genomics Viewer (IGV) interface. Alignment shows the location of the breakpoints and the 2-bp microhomology (CT) at the junction. The green and orange lines indicate part of the junction region originating from the proximal and distal sides of the deletion, respectively. The blue line indicates the 2-bp microhomology present at both sides of the deleted region. (**C**) Sanger sequencing electropherogram of the junction fragments for index patient 1[III:3]. The green, blue, and orange lines correspond to the lines described in Figure 2B. (**D**) Agilent DNA 12,000 Assay gel image of multiplex PCR for segregation analysis. Gel shows a 9612-bp fragment for the reference allele in all family members and an additional 13,232-bp fragment for the deleted allele in all affected family members and one family member with unknown disease status harboring the heterozygous *CRIM1* deletion.

**Figure 3 genes-12-00065-f003:**
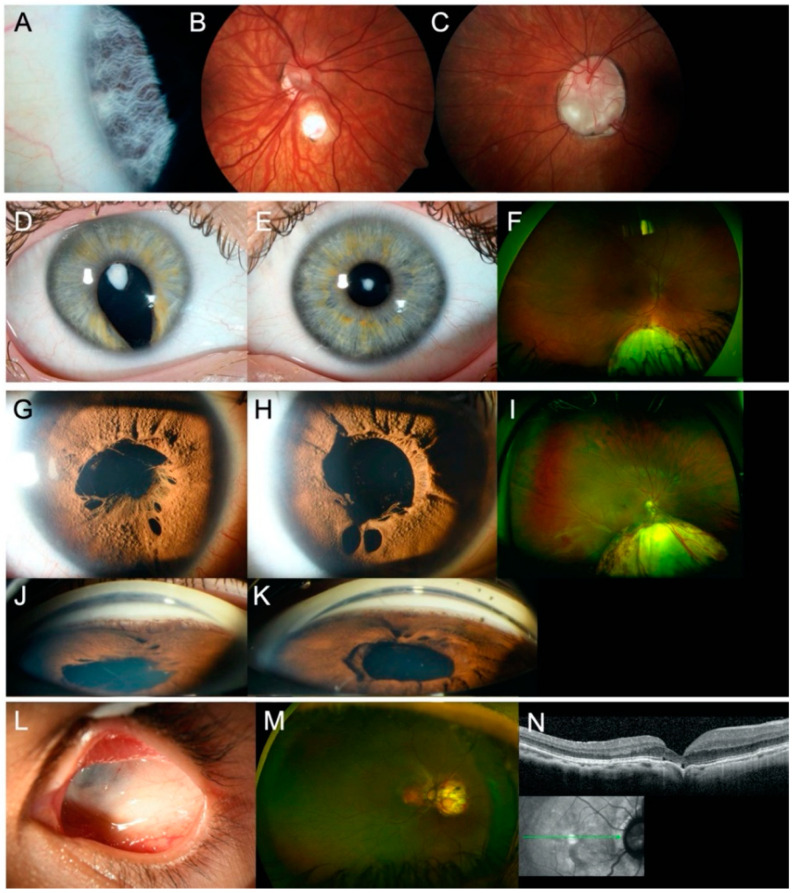
Ocular findings in affected individuals of families 2–4 and 7. (**A**–**C**) Slit lamp and fundus photographs of index patient 2[II:1] showing (**A**) posterior embryotoxon and (**B**) small chorioretinal coloboma in the right eye and (**C**) optic disc coloboma in the left eye. (**D**–**F**) Slit lamp photographs and Optomap™ image of index patient 3[II:1] showing (**D**–**E**) anterior polar cataract in both eyes and (**D,F**) iris and chorioretinal coloboma in the right eye. (**G**–**K**) Slit lamp photographs and Optomap™ image of index patient 4[II:2] showing Axenfeld-Rieger anomaly in both eyes, with (**G**–**H**) iris hypoplasia, polycoria, persistent pupillary membrane, (**I**) chorioretinal coloboma in the right eye (left eye not shown), and (**J**–**K**) goniodysgenesis in both eyes. (**L**–**N**) Slit lamp photograph, Optomap™ image, and spectral domain optical coherence tomography image of index patient 7[II:1] showing (**L**) severe microphthalmia in the left eye and (**M**–**N**) juxtapapillary chorioretinal coloboma in the right eye.

**Figure 4 genes-12-00065-f004:**
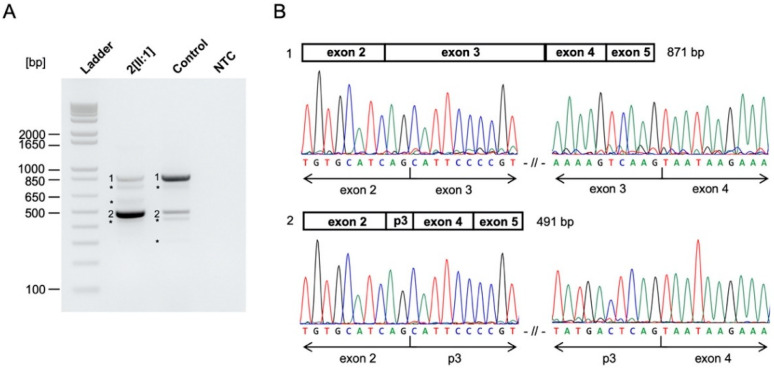
*CHD7* c.2095A>G variant alters splicing in functional analysis. (**A**) Agarose gel of RT-PCR products for patient- and control-derived cDNA and the NTC. RT-PCR showed correctly spliced transcript (Fragment 1), partial exon 3 skipping (Fragment 2), and additional nonspecific or alternatively spliced DNA fragments (not sequenced) in index patient 2[II:1] and the control. Partial exon 3 skipping (Fragment 2) was more pronounced in index patient 2[II:1] compared to the control. Ladder, 1 Kb Plus DNA Ladder (Thermo Fisher Scientific); NTC, no template control. * nonspecific or alternatively spliced DNA fragments. (**B**) Schematic overview and Sanger sequencing results of RT-PCR products. Sanger sequencing revealed correctly spliced *CHD7* transcript (Fragment 1) and a 380-nt deletion from the 3′ end of exon 3 (Fragment 2) due to the *CHD7* c.2095A>G variant. p3, partial exon 3.

**Figure 5 genes-12-00065-f005:**
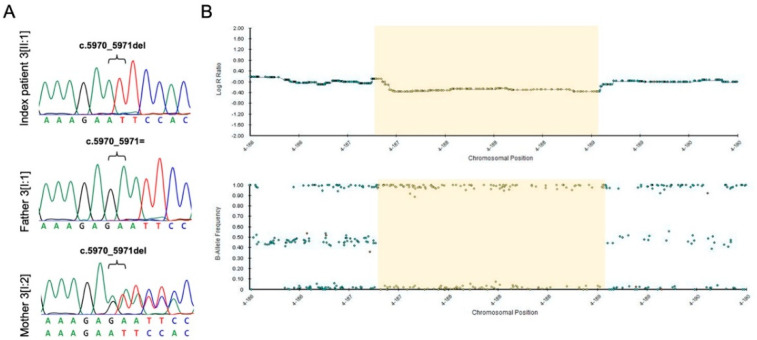
*FAT1* frameshift variant and deletion in index patient 3[II:1] and both parents. (**A**) Sanger sequencing results of index patient 3[II:1] and both parents for the *FAT1* frameshift variant. Index patient 3[II:1] and the mother (individual 3[I:2]) are hemizygous and heterozygous, respectively, for the *FAT1* frameshift variant (c.5970_5971del; p.(Asn1991PhefsTer19)). (**B**) Single nucleotide polymorphism (SNP) microarray data (B-allele frequency and Log R ratio chart) of the 4q35.1–q35.2 region for SNP array analysis of index patient 3[II:1]. Index patient 3[II:1] and the father (individual 3[I:1]) are hemizygous for SNPs in the deleted region (B allele frequency chart) and show reduced signal intensity (Log R ratio chart). The maximal deleted region is 1.8 Mb (chr4:187,149,541–188,971,489), with the approximately deleted region indicated in yellow.

**Table 1 genes-12-00065-t001:** Clinical details of index patients, affected family members, and asymptomatic carriers.

ID ^†^	Sex	Age (yrs) at Examination	Origin ^‡^	VA	Coloboma		Microphthalmia	Additional Ocular Anomalies	Extraocular Phenotype
					OD	OS			
1[III:3]	f	2	Portugal, Poland	0.5/0.25	NA	I	NA	None	None
1[II:3]	m	45	Portugal	0.1/NLP	IRCh	IRCh	NA	Microcornea (OU)	None
1[I:1]	m	75	Portugal	NA	I	NA	NA	Microcornea (OD)	None
2[II:1]	m	14	Netherlands	0.6/0.6	RCh	RCh	NA	Megalocornea (OD)	SNHL, DORV, VSD, PDA, DD, dysmorphic features
3[II:1]	f	17	Switzerland	0.2/0.6	IRCh	NA	NA	Anterior polar cataract (OU)	Syndactyly, hearing impairment
4[II:2]	f	30	Italy	NLP/0.8	RCh	RCh	NA	Axenfeld-Rieger anomaly (OU)	None
4[III:1]	f	0.75	Italy	0.25/0.25	I	IRCh	NA	None	VSD, clinodactyly
4[II:3]	f	24	Italy	1.0/1.0	NA	NA	NA	Goniodysgenesis (OU)	None
4[I:2]	f	56	Italy	0.6./0.8	NA	NA	NA	None	Uterine fibroids, keratocystic lesions
5[II:1]	f	7	Switzerland, Germany	NLP/0.5	IRCh	RCh	OD	None	DD, ASD, short stature
6[II:1]	m	2	Portugal	0.16/0.1	IRCh	IRCh	OS	None	None
7[II:1]	f	13	India	0.5/NLP	RCh ^a^	NA	OS	None	Aortic root enlargement
8[II:1]	f	12	Spain	0.6/1.0	I	I	NA	Axenfeld-Rieger spectrum (OU)	Tooth displacement
9[II:2]	f	33	Switzerland	0.1/0.1	RCh	RCh	OU	None	None
9[III:1]	m	6	Switzerland	LP/NLP	RCh	RCh	OU	None	None
10[II:1]	f	3	Switzerland, Germany	0.16/0.06	IRCh	IRCh	OS	None	None
11[II:2]	f	4	Switzerland, Belgium	0.6/0.8	I	I	NA	Cataracta corticonuclearis (initial partial inferonasal, at age 4 years complete; OD)	None
12[II:2]	f	2	Switzerland	0.05/0.08	IRCh	IRCh	NA	None	None
13[IV:4]	f	9	Switzerland, Austria	0.05/NLP	IRCh	NA	OS	None	Clinodactyly, mild pigeon toe, mild protruding ears
14[II:1]	m	19	Switzerland	0.4/0.05	RCh	RCh	NA	None	None
15[II:1]	f	13	Italy	0.4/0.8	RCh	RCh	NA	None	None

Abbreviations: ASD, atrial septal defect; C, coloboma; Ch, choroid; DD, developmental delay; DORV, double outlet right ventricle; f, female; I, iris; ID, intellectual disability; LP, light perception; m, male; M, microphthalmia; NA, not applicable; NLP, no light perception; OD, oculus dexter; OS, oculus sinister; OU, oculus uterque; PDA, patent ductus arteriosus; R, retina; SNHL, sensorineural hearing loss; VA, visual acuity; VSD, ventricular septal defect; yrs, years. ^†^ First numeral represents the family number; roman numeral represents the generational affiliation of the patient. ^‡^ Geographic origin of ancestors (three generations).^a^ Lateral coloboma.

**Table 2 genes-12-00065-t002:** Potentially disease-causing variants in index patients identified by whole-exome sequencing (WES).

ID ^†^	Gene	Gene Function	Reference Sequence	Sequence Variant (hg19)	Predicted Protein Change	Region/Size	gnomAD	Zygosity	ACMG	CADD	First Report
1[III:3]	*CRIM1*	Tether for growth factors, complexes with ß-catenin and cadherins	hg19	g.36769283_36778290del ^a^	NA	9,008 bp	NA	het	NA	NA	This study
2[II:1]	*CHD7*	Chromatin remodeling	NM_017780.3	c.2095A>G ^b^	p.Ser699Gly ^e^	exon 3	0%	het	vus	15.4	[26,31,32,33]
3[II:1]	*FAT1*	Cell polarity, cell migration, cell–cell adhesion	NM_005245.3	c.5970_5971del^c^	p.Asn1991PhefsTer19	exon 10	0%	hemi	vus	NA	This study
	*FAT1*, *F11*, *MTNR1A*, *ZFP42*		hg19	g.(187179210_187179486_188926200_189012426)del ^a^	NA	~1.8 Mb	NA	het	NA	NA	This study
4[II:2]	*PTCH1*	Hedgehog receptor	NM_000264.4	c.490G>A ^c^	p.Glu164Lys	exon 3	0%	het	vus	18.1	This study
5[II:1]	*PUF60*	Transcriptional regulation, pre-mRNA splicing, apoptosis	NM_001136033.2	c.752dup ^b^	p.Gln252ProfsTer152	exon 9	0%	het	P	NA	This study
6[II:1]	*BRPF1*	Chromatin regulator	NM_001003694.1	c.1756_1757insT ^b^	p.Glu586ValfsTer12	exon 5	0%	het	P	NA	This study
7[II:1]	*TGFB2*	Growth factor	NM_001135599.3	c.1043G>A ^d^	p.Arg348His	exon 7	0.00082%	het	LP	35	This study

Abbreviations: ACMG, American College of Medical Genetics and Genomics; CADD, Combined Annotation Dependent Depletion; het, heterozygous; hemi, hemizygous; NA, not applicable; LP, likely pathogenic; P, pathogenic; vus, variant of uncertain significance. † First numeral represents the family number, roman numeral represents the generational affiliation of the index patient. ^a^ Paternal segregation. ^b^ De novo. ^c^ Maternal segregation. ^d^ Segregation not known. ^e^ Splicing affected by variant, as demonstrated by functional analysis.

**Table 3 genes-12-00065-t003:** Additional variants identified by WES.

ID ^†^	Sex	Gene	Reference Sequence	Sequence Variant	Predicted Protein Change	Region	gnomAD	Zygosity	ACMG	CADD	Segregation
1[III:3]	f	*CRIM1*	NM_016441.2	c.926C>T	p.Pro309Leu	exon 5	0%	het	LB	25	paternal
1[II:3]	m										paternal
1[I:1]	m										NA
1[III:1]	f	*BRPF1*	NM_001003694.1	c.1489G>A	p.Ala497Thr	exon 3	0%	het	vus	25	maternal
3[II:1]	f	*TBX5*	NM_000192.3	c.349G>T	p.Ala117Ser	exon 4	0%	het	vus	24.7	maternal
		*FZD7*	NM_003507.1	c.1154C>T	p.Ala385Val	exon 1	0.0004%	het	vus	32	maternal
4[II:2]	f	*PPP1R12A*	NM_001143885.1	c.2014C>G	p.Pro672Ala	exon 15	0%	het	vus	24	paternal
4[III:1]	f										maternal
7[II:1]	f	*ACTG1*	NM_001199954.1	c.803-18dup	p.?	intron 4	0%	het	vus	NA	NA
8[II:1]	f	*EFTUD2*	NM_001142605.1	c.765-15C>G	p.?	intron 10	0%	het	vus	NA	paternal

Abbreviations: ACMG, American College of Medical Genetics and Genomics; CADD, Combined Annotation Dependent Depletion; f, female; het, heterozygous; m, male; NA, not applicable; vus, variant of uncertain significance. ^†^ First numeral represents the family number; roman numeral represents the generational affiliation of the patient.

## Data Availability

The data presented in this study are available on request from the corresponding author. WES data are not publicly available due to data protection regulations.

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
