# Peer review of "Whole Exome Sequencing in Coloboma/Microphthalmia: Identification of Novel and Recurrent Variants in Seven Genes"

_genes, 2021, doi:10.3390/genes12010065_

Round 1

Reviewer 1 Report

The authors present the genetic analysis of 15 unrelated families with Coloboma/Microphthalmia (C/M), using whole exome sequencing strategy and focusing on 307 genes previously related to these congenital eye malformations. Potentially pathogenic variants have been identified in 47% of the families, all but one reported for the first time. This study emphasizes the high genetic heterogeneity related to C/M and shows the complexity of establishing a molecular diagnosis in patients with C/M.

General comments:

  1. It’s really confusing changing all the time when referring to the cohort, mentioning “patients” (all the affected individuals enrolled in the study) or “index patients”, indistinctly. It would be more clear refer always to the index patients when presenting general results. The same recommendation should be taking into account when a subgroup of patients is referred.
  2. In the Results section, clinical findings are also presented, thus the subtitles (3.1., 3.2., …) must refer to both topics, genetic and ophthalmological.
  3. The 3.3. Additional Sequence Variants section is difficult to understand. The related families are not specifically presented and the results are only understandable when Table S2 is consulted, and this is a supplementary table.
  4. Figures should be numbered in the order they appear in the main text.
  5. In the Discussion section, whole genome sequencing should also be considered as a powerful and robust screening strategy.

Specific comments:

  1. (line 90) It is not necessary to indicate the particular author, there is a specific section to determine author contributions.
  2. (line 151) “…from family 1” must be added
  3. (line 199) Is it a subtitle?
  4. In figure 1, the particular mutation for each family should be cited in each title (under the family number and the altered gene), and then indicate each individual status using the symbols “+”/”m” (m1/m2 for the recessive case). Divorce is unnecessary to be shown.
  5. (lines 231 and 241) daughter is the individual 1[III:2]
  6. (line 235, figure 3, …) there is an error when describing the mutation. I suppose that the deletion comprehends the 3’ UTR, not the 5’ UTR.
  7. Table 1, Abbreviations commonly used in ophthalmology: OD, OS, OU (right eye, left eye, both eyes).
  8. (line 348) patient’s sister is the individual 4[II:3]
  9. (line 373) p.(Glu586Valfs*12)
  10. (lines 380, 572, 584) p.(Arg348His)
  11. Table S1, Genes should be arranged alphabetically.

Author Response

We are very grateful to the reviewer for his/her helpful comments and suggestions and have revised the manuscript accordingly. 

Reviewer 2 Report

Haug et al. evaluated 307 candidate genes over a set of 19 patients with coloboma or microphthalmia from 15 unrelated families. They report on the identification of seven novel and one recurrent variations that they assert are potential disease-causing mutations. They include a detailed clinical description of the phenotypes for each of the family members presented.

The criteria used to discriminate potential disease-causing mutations from the hundreds to thousands of other variations is sufficient to remove the obvious variants that are not disease-causing. However, it must be noted that these methods are insufficient to determine pathogenicity of the variants. Prediction algorithms are suggestive, but certainly not definitive, with numerous false-positives. Therefore, the variations included as potential disease-causing mutations are consistent with being disease-causing. And the authors have made strong arguments for each of them. But given the large number (307) of genes evaluated, there is a high likelihood that one or more of these are not the bona fide cause of disease.

Without additional families segregating the same or similar variations, or functional testing, it is not possible to determine if the majority of the potentially disease-causing mutations are legitimately disease-causing. Additional families and/or family members would be needed to discriminate bona fide DCMs from chance variations. Consistent segregation in one small family is insufficient. The fact that this study is limited by patient resources is a fact that should be stated - it's the reality that all rare-disease researchers face.

I found one error in this manuscript. On lines 235, the deletion in CRIM1 is described as spanning the last few exons and the 5' UTR. That is incorrect, it includes the 3' UTR. This same error is on line 273.

Author Response

We are grateful to the reviewer for his/her helpful comments and suggestions and have revised our manuscript accordingly. 

Reviewer 3 Report

In the present paper "Whole Exome Sequencing in Coloboma/Microphthalmia: Identification of Novel and Recurrent Variants in Seven Genes" the authors show an interesting research focused on the analysis of the DNA sequence of 19 patients with coloboma and/or microphthalmia, including family members.

The authors reported seven new variants and one recurrent one in genes related to this disease, CRIM1, CHD7, PTCH1, PUF60, and specially, BRPF1, FAT1 and TGFB2 as a novelty.

The study is stimulating and well presented.

Author Response

We are grateful to the reviewer for his/her comments and evaluation.

Reviewer 4 Report

The manuscript by Haug et al. describes the genetic analysis of patients with the ocular defects of coloboma and microphthalmia, as well as some associated family members. These conditions are heterogeneous, and genetic causality is unknown in the majority (up to 80%) of documented cases. The analysis was conducted using whole exome sequencing and subsequent filtering of the data using a compiled list of candidate disease genes previously associated with similar ocular phenotypes. Additional filtering methods and confirmation of the resulting sequence variants by segregation analysis, copy number assessment, and Sanger sequencing revealed seven novel putative disease-causing variants in six genes and one previously reported variant in another gene. Examination of additional phenotypes associated with alterations in these genes identified non-ocular pathology that can be monitored and potentially subjected to early treatment.

This is an important study that yields novel and significant information for the potential diagnosis and understanding of disease mechanisms underlying under-investigated ocular conditions. The data collection and evaluation was thorough and the resulting conclusions are convincing. This report will advance the field of coloboma and microphthalmia, and will serve as a valuable resource for other investigators and clinicians.

Minor editing of the manuscript is recommended as follows:

The notation used for numbers given in the text that are greater than one thousand often, and incorrectly, utilizes apostrophes (') rather than commas (,) - refer to lines 53, 152, 236, 239, 260, 281, and 343.

In line 175, there should not be an apostrophe (or comma) inserted in the name of the kit, "Agilent DNA 12000 Assay".

The 'Conclusion' section is quite extensive and spans nearly 5 pages. It would be beneficial if this section could be subdivided into sections (e.g. 4.1, 4.2, etc.) for improved organization and clarity.

Author Response

(The authors gave the same response as above.)
